# Signaling Switching from Hedgehog-GLI to MAPK Signaling Potentially Serves as a Compensatory Mechanism in Melanoma Cell Lines Resistant to GANT-61

**DOI:** 10.3390/biomedicines11051353

**Published:** 2023-05-03

**Authors:** Nikolina Piteša, Matea Kurtović, Nenad Bartoniček, Danai S. Gkotsi, Josipa Čonkaš, Tina Petrić, Vesna Musani, Petar Ozretić, Natalia A. Riobo-Del Galdo, Maja Sabol

**Affiliations:** 1Ruđer Bošković Institute, Division of Molecular Medicine, 10 000 Zagreb, Croatia; nikolina.pitesa@irb.hr (N.P.); matea.kurtovic@irb.hr (M.K.); josipa.conkas@irb.hr (J.Č.); tina.petric@irb.hr (T.P.); vmusani@irb.hr (V.M.); pozretic@irb.hr (P.O.); 2The Garvan Institute of Medical Research, Genome Informatics, Genomics & Epigenetics Division, 384 Victoria St., Darlinghurst, NSW 2010, Australia; nbartonicek@gmail.com; 3The Kinghorn Centre for Clinical Genomics, 370 Victoria St., Darlinghurst, NSW 2010, Australia; 4School of Molecular and Cellular Biology, Faculty of Biological Sciences, University of Leeds, Leeds LS2 9JT, UK; d.gkotsi@leeds.ac.uk (D.S.G.); n.a.riobo-delgaldo@leeds.ac.uk (N.A.R.-D.G.); 5Astbury Centre for Molecular Structural Biology, University of Leeds, Leeds LS2 9JT, UK; 6Leeds Institute of Medical Research, Faculty of Medicine and Health, University of Leeds, Leeds LS2 9JT, UK; 7Leeds Cancer Research Centre, University of Leeds, Leeds LS2 9JT, UK

**Keywords:** melanoma, HH-GLI pathway, RAS/RAF/ERK pathway, MAPK signaling, GANT-61, chemoresistance, primary cilia

## Abstract

Background: Melanoma represents the deadliest skin cancer due to its cell plasticity which results in high metastatic potential and chemoresistance. Melanomas frequently develop resistance to targeted therapy; therefore, new combination therapy strategies are required. Non-canonical signaling interactions between HH-GLI and RAS/RAF/ERK signaling were identified as one of the drivers of melanoma pathogenesis. Therefore, we decided to investigate the importance of these non-canonical interactions in chemoresistance, and examine the potential for HH-GLI and RAS/RAF/ERK combined therapy. Methods: We established two melanoma cell lines resistant to the GLI inhibitor, GANT-61, and characterized their response to other HH-GLI and RAS/RAF/ERK inhibitors. Results: We successfully established two melanoma cell lines resistant to GANT-61. Both cell lines showed HH-GLI signaling downregulation and increased invasive cell properties like migration potential, colony forming capacity, and EMT. However, they differed in MAPK signaling activity, cell cycle regulation, and primary cilia formation, suggesting different potential mechanisms responsible for resistance occurrence. Conclusions: Our study provides the first ever insights into cell lines resistant to GANT-61 and shows potential mechanisms connected to HH-GLI and MAPK signaling which may represent new hot spots for noncanonical signaling interactions.

## 1. Introduction

Among skin cancers, melanoma accounts for only 1% of all cases, but, due to its extremely high malignant properties, it is responsible for the majority of skin cancer-related deaths. In the last decade, great progress has been made and many important molecular mechanisms responsible for these malignant properties have been described. Despite that fact, patients suffering from malignant melanoma have a 10% five-year survival rate, highlighting melanoma as a serious public health problem [1]. Targeted therapy against components of the RAS/RAF/ERK signaling pathway is the most commonly used treatment for melanoma [2]. Almost 70% of all melanomas harbor BRAF^V600E^ or NRAS^Q61R^ mutations which lead to constitutive activation of the RAS/RAF/ERK signaling and, consequently, the promotion of cell proliferation, survival, invasion, and angiogenesis [3]. Specific RAS/RAF/ERK inhibitors have shown initial success in the clinic but, due to alternative reactivation mechanisms, like adaptive *RAS/RAF* mutations, RAF protein heterodimerization, receptor tyrosine kinase overexpression, loss of function mutations of negative regulators, and activation of other signaling pathways, therapy resistance frequently occurs [4]. In addition to that, many studies have demonstrated that MAPK signaling upregulation can mediate chemoresistance to inhibitors targeting other signaling pathways [5,6]. Hence, the combined targeting of RAS/RAF/ERK and other signaling pathways has received increased attention in the last years.

Hedgehog-GLI (HH-GLI) signaling has been implicated in a variety of cancers, including melanoma. Besides its canonical activation which is mediated by Hedgehog (HH) ligand binding, there is also known evidence of non-canonical crosstalk between HH-GLI and other oncogenic pathways [7]. RAS/RAF/ERK signaling promotes the proliferation and survival of melanoma cells by regulating the nuclear localization and transcriptional activity of the GLI1 transcription factor [8]. FDA-approved HH-GLI inhibitors primarily target upstream signaling components like Smoothened (SMO) but, unfortunately, adverse side effects, onset of SMO drug-resistant mutations, and noncanonical activation of the pathway limit their use [9]. A current strategy to overcome these limitations is the development of inhibitors of downstream signaling components like the GLI transcription factors. In comparison to the upstream antagonists, downstream inhibition of HH-GLI signaling demonstrated by experimental GLI antagonist, GANT-61, has shown to be most efficient in attenuating melanoma cell viability [10]. Targeting GLI1 and GLI2 transcription factors with GANT-61 restored sensitivity to vemurafenib-resistant cells, additionally confirming the importance of the non-canonical interplay between HH-GLI and RAS/RAF/ERK in melanoma [11]. 

In this study, we aimed to further investigate the interaction of MAPK signaling, in particular of the RAS/RAF/ERK module, and HH-GLI in chemoresistance. We established two melanoma cell lines with different *NRAS* mutational backgrounds resistant to GANT-61. For the first time, we report a characterization of established cell lines resistant to GANT-61 and identify potential mechanisms responsible for the development of resistance, which include changes in HH-GLI and MAPK signaling activity, cell cycle regulation, primary cilia formation, and elevated invasive properties. 

## 2. Materials and Methods

### 2.1. Generation of GANT-61 Resistant Cell Lines

Human melanoma cell lines derived from a primary tumor (Mel 224; RRID: CVCL U915) and pleural effusion metastasis (CHL-1; RRID: CVCL_1122) were kindly provided by Dr Neda Slade (Ruđer Bošković Institute, Zagreb, Croatia). The Mel 224 cell line harbors a homozygous NRAS^Q61R^ mutation, while the CHL-1 cell line is wild-type for both- *BRAF* and *NRAS* genes. Both cell lines were maintained in the recommended medium: CHL-1 in Dulbecco’s Modified Eagle Medium (Merck KgaA, Darmstadt, Germany); Mel 224 in RPMI 1640 medium (Merck KgaA, Darmstadt, Germany), both supplemented with 10% FBS (Merck KgaA, Darmstadt, Germany) and 1 mM sodium pyruvate; 1% streptomycin/penicillin; and 4 mM L-glutamine (Gibco Thermo Fisher Scientific, Waltham, MA, USA). To obtain resistant cell lines, cells were plated at low confluence (30–40%) and cultured for 9–12 months in the increasing concentrations of GANT-61 (Selleck Chemicals, Houston, TX, USA), starting from 1 μM and ending at 20 μM for CHL-1 or 30 μM for Mel 224. After the cells obtained resistance to GANT-61, they were maintained in a culture medium with 20 μM GANT-61 to prevent the loss of resistance.

### 2.2. MTT Viability Assay

In order to determine cell viability, compound 3-(4,5-Dimethylthiazol-2-yl)-2,5-diphenyltetrazolium bromide (MTT) was used as previously described [12]. Cells were plated in 96-well plates at 2 × 10^3^ cells/well. At 24 h post-plating, cells were treated with HH-GLI and RAS/RAF/MAPK pathway inhibitors—GANT-61 2.5–30 μM (GLI antagonist, Selleck Chemicals, Houston, TX, USA); cyclopamine (CYC, SMO antagonist) 1.25–10 nM (Selleck Chemicals, Houston, TX, USA); vismodegib (VDG, SMO antagonist) 1–50 μM (Selleck Chemicals, Houston, TX, USA); sonidegib (SDG, SMO antagonist) 1–50 μM (Selleck Chemicals, Houston, TX, USA); lithium chloride (LiCl, GSK3ß antagonist) 1–40 mM (Kemika, Zagreb, Croatia); arsenic trioxide (ATO, GLI antagonist) 0.03–125 μM (Sigma-Aldrich, St. Louis, MI, USA, SAD); and salirasib (SAL, RAS antagonist) 1.6–200 μM (Sigma-Aldrich, St. Louis, MI, USA, SAD)—for 72 h. Absorbance was measured on the LabsSystems Multiskan MS microplate reader (Thermo Fisher Scientific, Waltham, MA, USA) at 570 nm. The treatment was carried out in quadruplicate for each dose, and the experiment was repeated twice.

### 2.3. Quantitative Real-Time Polymerase Chain Reaction (qRT-PCR) 

Total RNA was extracted from the parental and resistant cell lines following TRIzol Reagent (Invitrogen, Waltham, MA, USA) protocol. cDNA was generated from 1 μg of RNA using the High-Capacity cDNA synthesis kit (Thermo Fisher Scientific, Waltham, MA, USA) and qRT-PCR performed on the CFX-96 instrument (Bio-Rad Laboratories, Hercules, CA, USA) using SsoAdvanced SYBR Green Supermix (Bio-Rad Laboratories, Hercules, CA, USA). The PCR conditions were as follows: initial denaturation at 95 °C for 3 min; 40 cycles of 95 °C for 15 s; 61 °C for 1 min; and finally melting curve from 70 °C to 95 °C. The results were analyzed with the CFX Manager Software v3.1 (Bio-Rad Laboratories, Hercules, CA, USA) normalized to the housekeeping gene *RPLP0,* and fold change was calculated using the 2^−ΔΔCt^ method [13]. Primers’ sequences used for qPCR are listed in Appendix A. 

### 2.4. Western Blot

Total proteins were extracted by sonication in radioimmunoprecipitation assay RIPA buffer) containing protease and phosphatase inhibitors (Complete Mini Protease Inhibitor Cocktail Tablets and PhosSTOP Inhibitor Tablets for phosphatases, both Roche, Basel, Switzerland). Protein concentration was measured using the BCA (Bicinchoninic Acid) kit (Thermo Fisher Scientific, Waltham, MA, USA). Proteins (50 μg) were separated on 7% or 12% SDS-polyacrylamide gel and transferred to a nitrocellulose membrane (Amersham BioSciences, Little Chalfont, England, UK). The quality of transfer was determined with Naphthol Blue Black (Sigma-Aldrich, St. Louis, MI, USA, SAD) staining. Membranes were blocked with 5% milk-TBST (Tris-Buffered Saline, 0.1% Tween^®^ 20 Detergent) solution and incubated with primary antibodies overnight. Primary antibodies used in this study are listed in Appendix A. After overnight incubation, membranes were washed in TBST and incubated for 1 h with the appropriate secondary HRP-conjugated antibodies—anti-rabbit 1:6000 (554021, BD Pharmingen, San Jose, CA, USA) and anti-mouse 1:8000 (554002, BD Pharmingen). Proteins were visualized using SuperWest Signal Pico and Femto reagents (Thermo Fisher Scientific, Waltham, MA, USA) on the Uvitec Image Alliance Q9 mini instrument (Uvitec, Cambridge, England, UK).

### 2.5. Wound Healing Assay

To determine the migration potential, cells were plated in 24-well plates at 10^5^ cells/well and left for 24 h to attach. At 24 h post-seeding, cell culture medium was changed out for fresh medium without FBS to exclude the effect of cell proliferation during the assay. At 24 h after starvation, two scratches per well were made with a 10 µL pipette tip; cells were then washed with phosphate buffered saline (PBS) to remove detached cells and treated with 10 μM GANT-61, 2 μg/mL CYC, 0.5 μM ATO or 20 mM LiCl. Scratch images were taken immediately after washing with PBS 18 and 24 h post-scratch at the same location using the DinoEye AM7023 camera (Dino-Lite, Naarden, The Netherlands). Eight images were taken for each treatment, and the images were processed using the MRI Wound Healing Tool plugin for FIJI to calculate the wound area [14].

### 2.6. Colony Forming Assay

For the colony forming assay, 2 × 10^3^ cells/well were plated in a 6-well plate, left to attach for 24 h, and then treated with HH-GLI inhibitors: 0.25–10 μM GANT-61; 2.5–15 mM LiCl, 1–4 μg/mL CYC; or 0.03–0.976 μM ATO. Cells were kept in culture for 2 weeks to allow colony formation, with the compound containing media being changed twice per week. Cells were then washed with PBS, fixed with 4% paraformaldehyde, and stained with Crystal Violet (Sigma-Aldrich, St. Louis, MI, USA, SAD) to visualize the colonies. Colonies were photographed and analyzed with FIJI software.

### 2.7. Immunofluorescence

For primary cilia visualization, cell lines were seeded in 24-well plates 5 × 10^5^ cells/well. At 24 h post-seeding, the complete medium was removed and medium without FBS was added. Primary cilia formation was additionally induced with 2.5 μM SAG (SMO agonist, Selleck Chemicals, Houston, TX, USA). After 72 h, cells were fixed with 4% paraformaldehyde, permeabilized with 0.025% Triton-X100 in PBS, and blocked with blocking solution (Dako, Glostrup, Denmark) for 1 h. Primary antibodies were incubated overnight: mouse anti-acetylated α-tubulin 1:100 (32-2700, Zymed Laboratories, South San Francisco, CA, USA) and rabbit anti-RAB34 1:100 (27435-1-AP, ProteinTech, Rosemont, IL, USA). Cells were washed with PBS and incubated with a secondary antibody: anti-mouse AlexaFluor^®^ 594 1:100 (8890S, Cell Signaling Technologies, Danvers, MA, USA) and anti-rabbit AlexaFluor^®^ 488 1:100 (4412S, Cell Signaling Technologies, Danvers, MA, USA) for 1 h. After secondary antibody incubation, cells were incubated with DAPI solution in PBS (Sigma-Aldrich, St. Louis, MI, USA, SAD) and visualized in the EVOS FLoid imaging system (Invitrogen, Waltham, MA, USA).

### 2.8. Flow Cytometry

Cell cycle analysis was performed using the Muse^®^ Cell Cycle kit (Luminex, Austin, TX, USA, SAD, cat. No. MCH100106) on the Guava^®^ Muse^®^ Cell Analyzer (Luminex), Cells were plated in 6-well plates at 2 × 10^5^ cells per well for three conditions: (1) untreated cells; (2) 100 nM doxorubicin (DOXO) (Sigma-Aldrich, St. Louis, MI, USA, SAD); and (3) 20/25 μM GANT-61 treatment (Selleck Chemicals, Houston, TX, USA). Cells were maintained in medium containing inhibitors for 48 h, collected, fixed, and analyzed according to the manufacturer’s instructions. PI3K/MAPK signaling activity was examined using the Muse^®^ PI3K/MAPK Dual Pathway Activation Kit that quantifies pERK 1/2 and pAKT levels (Luminex, Austin, TX, USA, cat. No. MCH200108). Cells were plated in 6-well plates at 2 × 10^5^ cells per well for three conditions: (1) 0.01 mM insulin (Sigma-Aldrich, St. Louis, MI, USA, SAD), which ensures detectable basal MAPK/PI3K activity; (2) 0.01 mM U0126 (Sigma-Aldrich, St. Louis, MI, USA, SAD), which acts as an MEK 1/2 antagonist; and (3) 15 μM and 20/25 μM GANT-61 (Selleck Chemicals, Houston, TX, USA) treatments. Cells were collected, fixed, permeabilized 30 min after treatment, and analyzed according to the protocol.

### 2.9. Statistics

The normality of data distribution was tested using the D’Agostino–Pearson test. Numerical data were presented with mean ± standard deviation. An independent samples t-test was used for comparing gene expression between parental and resistant cell lines, wound healing, cell cycle, MAPK signaling activity, and primary cilia analysis. The Pearson correlation coefficient (r) was calculated for analysis of colony formation capacity. *p*-values < 0.05 were considered statistically significant. Statistical analysis was performed using MedCalc v19.1.6 (MedCalc Software Ltd., Ostend, Belgium).

## 3. Results

### 3.1. Establishment of GANT-61-Resistant Mel 224 and CHL-1 Melanoma Cell Lines 

Before the establishment of resistant cell lines, MTT viability assay was performed to examine the cytotoxic effect of GANT-61 on parental cell lines Mel 224 and CHL-1. The obtained results showed that both cell lines displayed a concentration-dependent decrease in cell viability, but the CHL-1 cell line was slightly more sensitive to GANT-61 treatment compared to the Mel 224 cell line (IC50 5.78 μM (SD = 0.371) for CHL-1 vs. 11.06 μM (SD = 0.3812) for Mel 224). To establish resistant cell lines, Mel 244 and CHL-1 were cultured in the presence of increasing concentrations of GANT-61, starting from 1 μM. As the cells adapted, treatment concentrations were increased every month until cells were proliferating normally under constant exposure to 20 μM GANT-61. To verify that the cell lines gained resistance to GANT-61, we repeated the MTT assay in the same manner as with the parental lines. Both cell lines exhibited resistance to high GANT-61 concentrations, far above 10 μM, which is considered an average effective dose in general (Figure 1A) [15]. The Mel 224 cells demonstrated a higher level of resistance than CHL-1. In comparison to the IC50 value of the parental cell line 11.06 μM, the resistant Mel 224 (Mel 224 R) cell line showed an IC50 shift to 29.71 μM (SD = 6.401). In the case of CHL-1, the IC50 value increased from 5.78 μM in the parental cell line to 13.88 μM (SD = 0.6106) in the resistant cell line (CHL-1 R). To further confirm that the cell lines developed resistance to GANT-61 rather than displaying drug tolerance, resistant cells were cultured for two weeks in the absence of GANT-61 followed by repetition of the cytotoxicity test in response to increasing GANT-61 concentrations. While both cell lines retained resistance to GANT-61, viability curves had a biphasic shape where both cell lines showed reduced viability at low GANT-61 concentrations but resistance to higher concentrations. This cytotoxicity profile suggests a heterogenous population in which a subset of Mel 224R and CHL-1 R cells was not stably resistant to GANT-61. In addition, the CHL-1 R cells showed an even bigger shift of IC50 value (24.62 μM) in comparison to the previous viability assay, which further confirmed our assumptions that the cells had acquired GANT-61 resistance. Colony formation assays showed that both resistant cell lines exhibited a higher colony formation capacity than the parental cell lines and that, while all cell lines were sensitive to GANT-61 in a dose-dependent manner, the resistant cells were affected at higher concentrations (Figure 1B). Surprisingly, in the CHL-1 R cells, GANT-61 treatment positively affected cell migration, increasing the migratory potential in comparison to untreated conditions. On the other hand, the Mel224 R cells were less migratory after the treatment with GANT-61 (Figure 1C).

### 3.2. Mel 224 and CHL-1 Cell Lines Resistant to GANT-61 Exhibit Morphological and Molecular Changes 

During the establishment of resistant cell lines, first, we noticed that the resistant cell lines changed their morphology in comparison to the parental cell lines (Appendix A). Both resistant cell lines showed a spindle-like-shaped cell phenotype which might be associated with epithelial–mesenchymal transition (EMT) and their increased migratory capacity. To investigate how resistance development affects HH-GLI signaling and cellular process which are under its control, we analyzed the mRNA expression of two gene panels, the first covering HH-GLI components *GLI1*, *GLI2*, *GLI3*, *PTCH1*, *SHH*, *SUFU*, *SMO* and *GSK3*B (Figure 2A), and the second which included genes involved in EMT and invasion (*VIM*, *CDH1*, *MMP2*, *MMP9*), cell proliferation (*C-MYC*, *KI67*), and stemness (*NANOG*, *OCT4*, *SOX2*) (Figure 2C). Both GANT-61 resistant cell lines showed upregulation of *GLI1* expression (Figure 2A). However, while Mel 224 R cells displayed a general downregulation of most other HH-GLI pathway components, the CHL-1 R cell line showed no change and additional upregulation of *PTCH1*, another GLI-target gene. Analysis of HH-GLI pathway components at the protein level showed that both cell lines exhibit downregulation of most components (Figure 2B). Unexpectedly, GLI1 protein level is reduced in Mel 224 R cells compared to parental cells, and slightly upregulated in CHL-1 R cells, suggesting changes in the post-transcriptional regulation of GLI1 after GANT-61 exposure. In agreement with the observed morphological and migration changes, both resistant cell lines had upregulated expression of the mesenchymal marker *VIM* and of *MMP2*, which is associated with invasive cell properties (Figure 2C). The Mel 224 R cells also showed downregulation of the epithelial marker *CDH1*, overall suggesting that GANT-61 resistance promotes a more mesenchymal phenotype than the parental cells. Despite those commonalities, each resistant cell line exhibited a distinct pattern of expression of stemness-associated transcription factors. *SOX2* and *NANOG* were upregulated and *OCT4* downregulated in the Mel 224 R cells, but changed in the opposite direction in the CHL-1 R cells (Figure 2C).

Given the known crosstalk between the HH-GLI and MAPK signaling and their involvement in melanoma development, we also examined the expression and activation status of ERK1/2, JNK, and p38 MAPKs at the protein level. The Mel 224 R cells showed a dramatic increase in the level of phospho-ERK and reduction of phospho-p38, while the CHL-1 R cells had an opposite pattern: increased phospho-p38 level and a reduction in ERK signaling (Figure 2D). The opposite changes in the ERK phosphorylation prompted us to examine the expression of the negative regulators, *DUSP4*, *DUSP7*, *SPRY2,* and *SPRY4* (Figure 2E). These negative regulators also represent potential GLI transcriptional targets which we have identified within our previous study [10]. Consistent with our previous results, *DUSP4*, *DUSP7,* and *SPRY4* were downregulated in the Mel 224 R cell line, while gene expression of *SPRY2*, *SPRY4,* and *DUSP4* was upregulated in the CHL-1 R cell line in comparison to the parental cell line, which could explain the observed changes in ERK activity (phosphorylation) levels. Both resistant cell lines exhibited slightly downregulated levels of phospho-JNK. Finally, we used a flow cytometry approach to investigate MAPK/PI3K signaling activity modulation by acute treatment with an MEK inhibitor (U0126) or GANT-61 in parental and resistant cells. The kit detects phospho-ERK1/2 and phospho-AKT positive populations and, therefore, identifies the activation of the MAPK, the PI3K, or both signaling pathways. For that reason, flow cytometry enables us to quantify trends that we previously observed with western blot analysis—namely, ERK signaling hyperactivation. The Mel 224 and Mel 224 R cell lines showed similar MAPK/PI3K activity profiles in basal conditions (red and teal column segments, Figure 2F). The U0126 treatment increased the dual MAPK/PI3K negative population in the Mel 224 R cells much more than in the Mel 224 cells (*p* = 0.0072) (light blue column segments). Interestingly, acute treatment with 10 mM and 20 mM GANT-61 increased the MAPK positive cell population in Mel 224 R compared to the Mel 224 cells (*p* = 0.001). The CHL-1 and CHL-1 R cell lines showed a similar response to U0126 and GANT-61 but not statistically significant: as we described for Mel 224 R, the CHL-1 R cells appeared more sensitive to U0126 treatment than CHL-1, and showed an increase in MAPK/PI3K negative cell population.

### 3.3. Differential Sensitivity of GANT-61 Resistant Cell Lines to HH-GLI and RAS Inhibitors

Once we successfully validated our resistant cell lines, we examined their response to other HH-GLI and RAS/RAF/MAPK inhibitors in terms of proliferation and invasive cell properties like colony formation capacity and migration. To determine the response to other inhibitors, we performed MTT viability assays as previously described. Both Mel 224 and Mel 224 R cell lines showed a dose-dependent reduction of viability in response to arsenic trioxide (ATO), a non-selective GLI protein inhibitor [16,17]; however, Mel 224 R cells exhibited lower sensitivity to low–mid concentrations of ATO in comparison to the parental cell line (Figure 3A). In comparison to GANT-61 treatment, where the resistance was maintained after two weeks without GANT-61 in the culture medium, here, the relative ATO resistance was lost, indicating that the cells transiently tolerated ATO treatment (Appendix A). The response to other inhibitors—LiCl (GSK3ß inhibitor), cyclopamine, vismodegib, sonidegib (CYC, VDG, SDG; SMO inhibitors), and salirasib (SAL; RAS inhibitor)—did not significantly change (Figure 3A and Appendix A). Among all tested inhibitors, only ATO and SDG completely decreased cell viability, while others were not that effective, and at least 20% of viable cells remained. In the CHL-1 R cell line, we did not detect any cross-resistance to any of the tested inhibitors in comparison to the parental cell line. Interestingly, in both cell lines, we detected a big difference in efficacy between clinically-approved SMO inhibitors VDG and SDG (Appendix A). SDG showed a stronger reduction of cell viability although both inhibitors have the same protein target and mechanism of action. Interestingly, our results are consistent with clinical outcomes for patients who have been treated with SMO inhibitors [18]. Colony formation assays showed that both resistant cell lines have a higher colony formation capacity than parental cell lines in the presence of HH-GLI inhibitors ATO, LiCl, and CYC (Figure 3B). In the case of the Mel 224 R cell line, there was a clear dose-dependent response to all three conditions—ATO (r = −0.08371, *p* = 0.0095), CYC (r = −0.9187, *p* = 0.0013), and LiCl (r = −0.769, *p* = 0.0257)—while, in the CHL-1 R cells, only LiCl (r = −0.8645, *p* = 0.0056) showed a significant dose-dependent decrease in colony formation capacity. 

Migration analysis showed that both resistant cell lines had a higher migration potential under basal conditions in comparison to the parental cell lines (24 h *p* = 0.0003 for Mel 224 R and 24 h *p* = 0.0027 for CHL-1 R) (Figure 3C). Overall, no inhibitor reduces the migratory potential in Mel 224, while the Mel 224 R cells are negatively affected after all tested inhibitors. Besides having a higher migration potential in untreated conditions, the CHL-1 R cell line also exhibited a higher migration potential in the presence of LiCl (18 h *p* = 0.0134) compared to CHL-1. In comparison to untreated conditions, LiCl was the most effective in attenuating migration potential in both CHL-1 cell lines (24 h *p* = 0.0179 for CHL-1 and 24 h *p* = 0.0001 for CHL-1 R).

### 3.4. The Influence of GANT-61 Resistance on Cell Cycle Regulation 

To investigate whether the established resistant cell lines exhibit changes in cell cycle regulation, we performed cell cycle analysis. In untreated conditions, the Mel 224 and Mel 224 R cell lines did not show significant differences in their cell cycle profiles (Figure 4A). Next, we treated the cells with doxorubicin (DOXO), an antitumor agent that induces double-stranded breaks and cell cycle arrest in the G2/M phase, and with GANT-61 that induces cell cycle arrest in G0/G1 [19,20]. DOXO caused the expected G2/M accumulation in the resistant cell line (*p* < 0.0001), while this effect was not observed in the parental cell line; however, a significant increase in the sub-G1 population which represents apoptotic cells was detected instead (*p* < 0.0001), (Figure 4B) [21]. We detected the same trend in the parental cell line after 25 μM GANT-61 treatment (*p* < 0.0001), while G0/G1 accumulation was observed in the resistant cell line (*p* = 0.0002). In all samples, regardless of the conditions, we could detect a multinucleated cell population, which can be identified as a signal after G2/M cell population during cell cycle analysis (Figure 4B,C and Appendix A). Multinucleated cells occur as a result of genomic instability and are very common in cancer. Many studies have shown that these cells can contribute to drug resistance [22]. DOXO and 25 μM GANT-61 treatment caused a significantly larger increase in multinucleated cell population in Mel 224 R in comparison to the Mel 224 cell line (*p* = 0.0001; *p* = 0.0005 respectively). In the CHL-1 cell line, cell cycle profiles differed between the parental and resistant cells. In untreated conditions, the CHL-1 R cell line had a higher percentage of G0/G1 cell population in comparison to CHL-1 (*p* = 0.0001). Contrary to the Mel 224 cell lines, DOXO treatment caused the expected G2/M accumulation only in the CHL-1 cell line (*p* < 0.0001), while the CHL-1 R cell line had a higher sub-G1 population percentage (*p* < 0.0001), indicating apoptosis. With GANT-61 treatment, we have observed the same effect—G0/G1 accumulation in CHL-1 (*p* = 0.0001) and increased sub-G1 population in CHL-1 R cell line (*p* = 0.0001) (Figure 4B). 

### 3.5. Primary Cilia Formation in Generated GANT-61 Resistant Cell Lines

In one of our previous studies, we identified novel GLI transcriptional targets using a combined RNA-*seq* and ChIP-*seq* approach. *RAB34*, as a novel GLI2 transcriptional target, is a crucial regulator of primary cilia formation [10]. Since canonical HH-GLI signal transduction is highly dependent on the properly formed primary cilium, we examined RAB34 expression on the established resistant cell lines. The Mel 224 R cell line exhibited lower RAB34 expression in comparison to the parental cell line at both transcript and protein levels, while the CHL-1 cell line did not show any differences (Figure 5A,B). Given the previous results, we wanted to further investigate whether RAB34 downregulation in the Mel 224 R cell line is associated with reduced primary cilia formation, using immunofluorescence against acetylated α-tubulin to visualize primary cilia. We detected low levels of primary cilia in both parental Mel 224 and Mel 224 R. Furthermore, we found a clear colocalization of acetylated α-tubulin staining and RAB34 in the primary cilia body (Figure 5C). We photographed each cell line in six visual fields and quantified the primary cilium incidence. In the Mel 224 cell line, we counted in total 1410 cells of which 64 had primary cilia (4.5% incidence), while Mel 224 R cells had an even lower incidence of ciliation (2.5%, 48/1945 cells). Thus, the resistant cell line exhibited a 54.37% reduction of primary cilia formation (*p* = 0.0308). The finding that the cells that form the primary cilia represent a small per cent of the total population is not a surprising observation since it is known that melanoma cells often show primary cilia loss, which can be connected with increased oncogenic properties [23,24]. Therefore, we still need to investigate in more depth whether further cilia reduction in the resistant cell line has a biological impact on cell properties associated with resistance establishment. In the CHL-1 cell line, we did not detect any primary cilia, although we previously observed RAB34 expression (Figure 5A,B,D). These findings suggest that RAB34 has other cell functions independent of ciliogenesis.

## 4. Discussion

The RAS/RAF/MAPK and HH-GLI signaling pathways are known to interact in melanoma and to promote tumorigenic properties like cell proliferation, survival, and invasiveness [8]. Chemoresistance presents a major challenge in melanoma therapy; hence, the effect of specific BRAF and MEK inhibitors in clinical use is limited and new therapeutical approaches are necessary. To this day, the RAS/RAF/MAPK and HH-GLI interplay has not been studied sufficiently in terms of chemoresistance. Several studies have confirmed the importance of RAS/RAF/MAPK and HH-GLI interaction in this process and have shown that HH-GLI inhibition, more specifically targeting the GLI transcription factors, can restore sensitivity to specific EGFR and BRAF inhibitors [11,25]. 

In this study, we present for the first time an in-depth characterization of two melanoma cell lines resistant to GANT-61, a specific GLI protein antagonist. The CHL-1 NRAS*^WT^* cell line was more sensitive to long-term GANT-61 treatment than the Mel 224 NRAS*^Q61R^* cell line, as it was harder to establish permanent resistance. These results are consistent with studies showing that the *NRAS* mutated melanoma is more aggressive, and associated with elevated mitotic activity and higher metastatic properties when compared to the *NRAS* wild-type melanoma [26]. However, a limitation of our study is the lack of BRAF*^V600E^* mutated melanoma lines, which will be investigated in the future. Our established cell lines displayed a great number of molecular and morphological changes, with some commonalities, but mainly showing different mechanisms underlying GANT-61 resistance. One of the observations in both resistant cell lines, which was distinctly noticeable, was the change in cell morphology. In comparison to the parental cell lines, the resistant cell lines had a spindle-like shape and were bigger in size. Our hypothesis that these changes might be associated with EMT was confirmed with the detection of downregulated gene levels of the epithelial marker *CDH1* and upregulated levels of the mesenchymal marker *VIM* in the resistant cell lines. It is known that cells undergoing EMT have increased invasive properties by producing more matrix metalloproteinases (MMPs) [27]. In our case, we observed upregulated *MMP2* gene levels in both resistant cell lines. Besides being involved in cell invasion, EMT-related transcription factors can trigger the expression of stemness factors [28]. Since cancer stem cells (CSCs) have the ability to self-renew and are less sensitive to therapy which makes them often responsible for resistance occurrence, we checked the gene expression levels of stemness-related transcription factors, *OCT4*, *NANOG,* and S*OX2* [29]. The Mel 224 R cell line exhibited upregulated *NANOG* and *SOX2* levels, while the CHL-1 R cell line showed upregulation of *OCT4*, suggesting increased stemness in both cell lines through different mechanisms. 

On the protein and mRNA levels, both resistant cell lines exhibited downregulation of the majority of the analyzed HH-GLI signaling components. Protein levels of all specific HH-GLI signaling components (GLI1, GLI2, GLI3, PTCH1, SMO) were downregulated in the resistant cell lines. In the case of GLI1 protein, we identified two isoforms, the 160 kDa full length protein isoform (GLI1FL) and a ~130 kDa isoform, possibly the GLI1ΔN which is lacking the N-terminal SUFU binding domain [30]. While both isoforms were downregulated in the Mel 224 R cell line, we detected an opposite expression pattern in the CHL-1 cell line, upregulation of GLI1FL in CHL-1 R and downregulation of GLI1ΔN in CHL-1. Although these isoforms have the same DNA binding domain, it is known that they have different mechanisms of target gene activation and transcriptional potency [31]. Therefore, it would be interesting to further investigate if resistance occurrence is potentially associated with changes in GLI1 splicing regulation in the CHL-1 cell line. GSK3ß kinase was upregulated on the protein level in Mel 224 R cells, which can indicate multiple adaptive mechanisms since GSK3ß regulates numerous other signaling pathways besides HH-GLI. There are many studies that have demonstrated the importance of GSK3ß in drug resistance maintenance [32].

It is known that tumors originate from multiple clones of malignant cells with different properties, which can be selected due to applied therapy [33]. Therefore, HH-GLI signaling downregulation detected here could potentially be a result of simple cell selection where cells with lower basal GLI protein expression gained selective advantage and survived prolonged GANT-61 treatment. A drug’s efficacy is influenced by its molecular target and alterations of this target, such as mutations or modifications of expression levels; therefore, there is also a possibility that cells altered their GLI protein expression to avoid sensitivity to GANT-61 treatment upon long-term therapy [34]. In that case, cancer cells can often activate compensating signaling loops which elicit similar phenotypic consequences as the original pathway to further bypass drug-mediated inhibition [35]. 

Interestingly, we observed that different MAPK signaling pathways were hyperactivated in the resistant cell lines. MAPK signaling is composed of three main cascade modules, ERK1/2, p38, and JNK, each of which has its own substrates and specific biological functions [36]. The ERK1/2 module, activated by RAS/RAF, is predominantly pro-proliferative, while JNK and p38 can have dual tissue-specific roles in cell proliferation, apoptosis, and inflammatory responses [37]. We observed increased ERK activation, reflected by increased phosphorylation, and p38 inhibition in the Mel 224 R cell line compared to its parental cells, while an opposite activation trend was observed in the CHL-1 R cell line, with increased p38 and decreased ERK phosphorylation. Irrespective of which module is upregulated, these signaling changes could present potential mechanisms associated with GANT-61 resistance. There are several studies which confirm that MAPK upregulation can mediate chemoresistance to inhibitors targeting other signaling pathways [5,6,38]. Therefore, MAPK activation in the resistant cell lines could pose a potential signaling alternative in conditions of downregulated HH-GLI signaling. Kuonen et al. have demonstrated a switch in signaling from HH-GLI to RAS/RAF/ERK in basal cell carcinoma cell lines resistant to the SMO inhibitor, vismodegib [39]. RAS/RAF/ERK upregulation was further verified with flow cytometry in the Mel 224 R cell line by demonstrating increased pERK levels after GANT-61 treatment and higher sensitivity to U0126 in terms of MAPK activity. Changes in RAS/RAF/ERK activity were accompanied with altered gene expression of ERK negative regulators *SPRY2*, *SPRY4*, *DUSP4,* and *DUSP7*. Consistent with our previous observations, *SPRY4*, *DUSP4,* and *DUSP7* were downregulated in Mel 224 R, while *SPRY2*, *SPRY4,* and *DUSP4* were upregulated in the CHL-1 R cell line. Balko et al. demonstrated that MAPK activation upon *DUSP4* loss promotes cancer stem cell-like properties in basal-like breast cancer cells [40]. Yao et al. showed that SPRY2 regulates proliferation and survival of multiple myeloma via inhibiting the activation of ERK1/2 pathway, while Kumar et al. identified SPRY4 as the potential mediator of growth suppression in melanoma upon dual BRAF^V600E^ and NRAS^Q61^ oncogene expression [41,42]. 

To characterize more thoroughly our established cell lines, we examined their response to other HH-GLI and RAS inhibitors. HH-GLI inhibitors can target upstream and downstream components of the signaling cascade. Cyclopamine, vismodegib, and sonidegib bind to SMO on the cell membrane and inhibit downstream signaling transduction [43,44]. Lithium chloride has been shown to inhibit canonical HH-GLI signaling by multiple ways: causing cilia elongation, increasing GLI-SUFU interaction by inhibition of GSK3b, and promoting GLI1 proteasomal degradation [12,45,46]. Arsenic trioxide antagonizes HH-GLI signaling through direct binding and inhibition of GLI proteins, but, in contrast to GANT-61 which was described as a selective inhibitor, arsenic trioxide has multiple substrates besides GLI proteins [16,17]. In comparison to the parental cell lines, the Mel 224 R and CHL-1 R cell lines did not show any differences in response to downstream inhibitors CYC, VDG, SDG, and LiCl, suggesting that they became independent of HH-GLI signaling. Our results demonstrated that CYC and VDG had no or very little effect on cell viability, confirming many studies which have shown that upstream HH-GLI inhibition is often ineffective, especially in cases of predominant noncanonical HH-GLI activation [9,10]. Since our resistant cell lines show downregulation of SMO, theoretically, we expected that they will perhaps exhibit increased sensitivity, or cross-resistance to SMO inhibitors, but we did not find any differences in response to SMO inhibitors between parental and resistant cell lines. LiCl reduced cell viability which is not surprising since GSK3ß kinase regulates multiple signaling pathways besides HH-GLI [47]. Interestingly, we observed a remarkable difference in efficacy between two FDA-approved SMO inhibitors, SDG and VDG, in all cell lines, where SDG showed to be more potent. These results seem to reflect clinical outcomes, although there are studies which observed opposite trends after treatment with SMO inhibitors [18,48]. The difference in therapeutic efficacy could potentially be explained by the study of Malhi et al. which demonstrated that VDG poses as a substrate for P-glycoprotein and is metabolized by cytochrome P450, 2C9, and 3A4 which can affect its pharmacokinetics [49]. On the other hand, only Mel 224 R exhibited transient cross-resistance to ATO. There are many studies pointing out the presence of persister cells population within the tumor, which plays a major role in therapy resistance. These cells do not harbor classic drug resistance driver alterations and their partial resistance phenotype is transient and reversible upon removal of the drug [50]. We believe that this effect is a result of therapy tolerance caused by the unselective traits of ATO which were mentioned earlier. Salirasib, a farnesyl transferase inhibitor that disrupts RAS protein localization to the cell membrane, was only effective at high doses, which can potentially be the result of unspecific general cytotoxic events [51]. Downstream RAS/RAF/ERK inhibitors like MEK and ERK antagonists would probably be more suitable for future studies than upstream inhibitors in Mel244 R cells, and inhibitors of p38 MAPK could be valuable in CHL-1 R cells.

Since we have previously detected upregulated *MMP2* gene levels in both resistant cell lines, we wanted to examine if they exhibit elevated invasive traits by analyzing cell migration and colony formation capacity. In comparison to the parental cell lines, Mel 224 R and CHL-1 R showed a higher migratory and colony formation capacity upon treatment with HH-GLI inhibitors, GANT-61, ATO, CYC, and LiCl. A study of Yang et al. showed that RAS/RAF/ERK signaling can mediate MMPs’ activity which influences cell migration and invasion [52]. Guided by observed upregulated gene levels of stemness-related transcription factors and known HH-GLI involvement in cell cycle regulation, we also examined if any changes in cell cycle regulation occurred in the resistant cell lines. In basal conditions, we did not observe any differences in cell cycle regulation between the Mel 224 and Mel 224 R cell lines; however, after DOXO and GANT-61 treatment, we detected a sub-G1 cell population in the Mel 224 cell line, which indicates apoptosis. Since the cell cycle profiles did not vary between the cell lines in basal conditions, we hypothesize that observed changes are not associated with resistance occurrence. On the contrary, the CHL-1 R cell line exhibited a higher percentage of cell population in sub-G1 in comparison to the parental cell line after treatment with DOXO or GANT-51. The CHL-1 R cell line also had a significant increase in G0/G1 cell population in basal conditions. Accumulation in the G0/G1 phase could indicate a slow-cycling cell state, which can be associated with invasion, chemoresistance, and stemness [53,54]. 

In a previous study, we identified numerous novel GLI transcriptional targets by combining RNA-*seq* and ChIP-*seq* [10]. The *RAB34* gene, a novel GLI2 transcriptional target, is required for fusion of preciliary vesicles and, consequently, for normal primary cilia formation. HH-GLI signaling is highly dependent on proper cilia formation and it has been shown that RAB34 activity is crucial for canonical HH-GLI signaling [55]. Since we previously detected downregulation of HH-GLI signaling in the resistant cell lines, we examined if RAB34 expression had changed. We did not observe any changes in RAB34 protein expression in the CHL-1 cell lines; however, RAB34 was downregulated in the Mel 224 R cell line. Intrigued by these results, we checked primary cilia formation capacity in our cell lines and observed a 54% reduction of primary cilia incidence in Mel 224 R in comparison to the parental cell line. Kuonen et al. have previously demonstrated primary cilia loss upon signaling switching from HH-GLI to RAS/RAF/ERK in basal carcinoma cell lines resistant to vismodegib [39]. A study of Radford et al. showed that upregulated RAS/RAF/ERK activity, specifically ERK1/2, leads to primary cilia shortening, while Zhang et al. identified RAB34 as a potential substrate for MAPK kinases [56,57]. Primary cilia loss often occurs in melanoma, and there are studies showing that cilia loss is associated with elevated invasive traits, poorer histopathological features, and therapy outcomes [58,59]. We did not detect primary cilia in the CHL-1 and CHL-1 R cell lines which might be due to other cell line characteristics like their mutation background and origin. Interestingly, primary cilia loss in the Mel 224 R cell line could potentially explain the downregulation of HH-GLI components beside GLI proteins like SMO and PTCH1, which are usually accumulating on the primary cilia membrane and are mandatory for canonical HH-GLI signaling. 

## 5. Conclusions

Here we report, for the first time, a comprehensive characterization of melanoma cell lines of different genetic backgrounds resistant to GANT-61. Although we established two cell lines resistant to the same drug, potential mechanisms involved in resistance occurrence differ between them. However, there are shared traits among them which need to be investigated further. Both cell lines exhibited HH-GLI signaling downregulation and switching to different modules of MAPK signaling, cancer stem-like properties and elevated invasive characteristics visible through increased migration, colony formation capacity, and EMT. The established cell lines did not show permanent cross-resistance to other tested HH-GLI and RAS inhibitors. Our study provides new insights into the HH-GLI and RAS/RAF/ERK interplay associated with ciliogenesis regulation via RAB34.

## Figures and Tables

**Figure 1 biomedicines-11-01353-f001:**
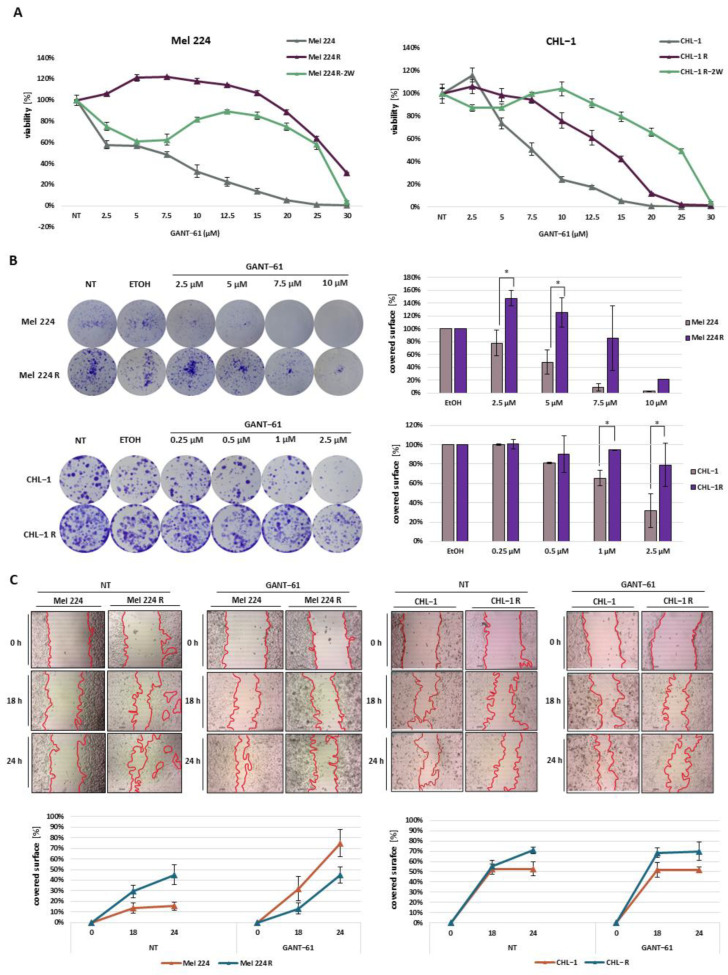
Validation of established cell lines resistant to GANT-61. (**A**) Cell viability of parental Mel 224 and CHL-1, resistant (R) cell lines after continuous treatment with GANT-61, and resistant (R-2W) cell lines two weeks after no treatment with GANT-61. (**B**) Colony formation capacity of Mel 224 and CHL-1 cell lines after GANT-61 treatment. (**C**) Migration capacity of established cell lines in untreated conditions (NT) and after GANT-61 treatment examined with wound healing assay. * denotes a statistically significant difference (*p* < 0.05).

**Figure 2 biomedicines-11-01353-f002:**
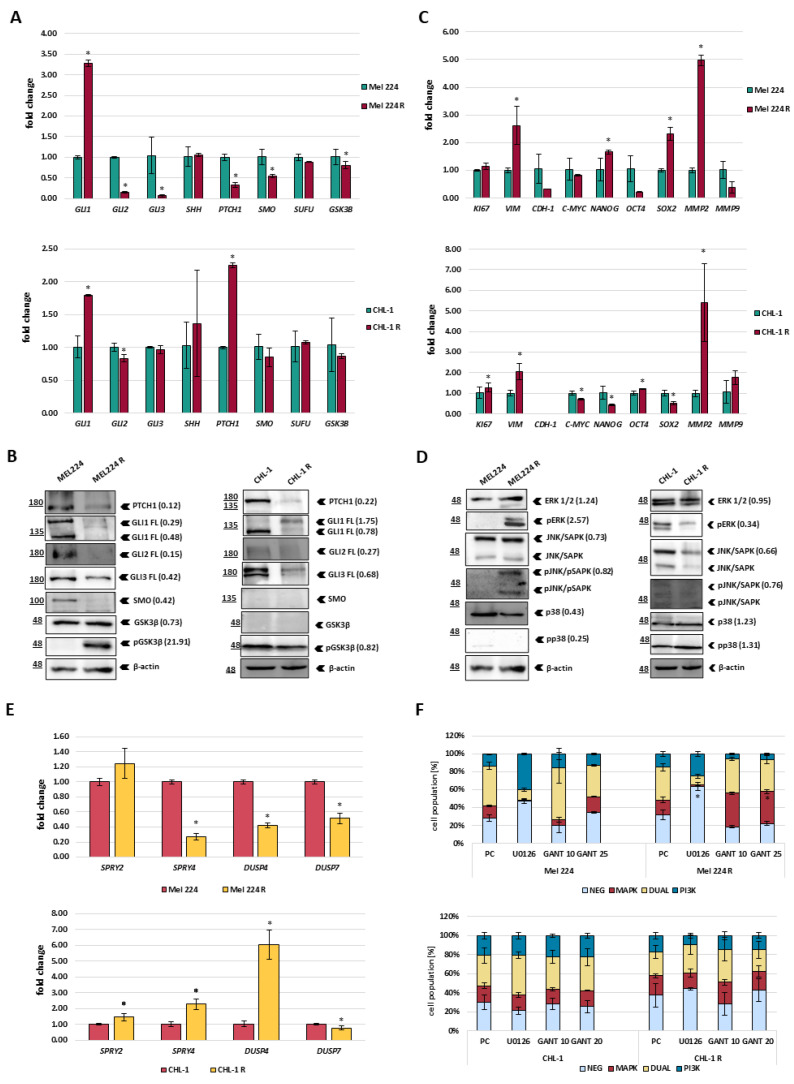
Molecular changes of established resistant cell lines. (**A**) Gene expression levels of HH-GLI components (*GLI1*, *GLI2*, *GLI3 PTCH1*, *SHH*, *SUFU*, *SMO,* and *GSK3B*) evaluated by qPCR. (**B**) Protein levels of HH-GLI (GLI1, GLI2, GLI3, PTCH1, SMO, and GSK3ß) evaluated by western blot. (**C**) Gene expression levels of EMT and invasion (*VIM*, *CDH-1*, *MMP2,* and *MMP9*), proliferation (*C-MYC* and *KI67*) and stemness markers (*NANOG*, *OCT4,* and *SOX2*) evaluated by qPCR. (**D**) Protein levels of MAPK components (ERK1/2, JNK, and p38) examined by western blot. (**E**) Gene expression of negative RAS/RAF/ERK regulators evaluated by qPCR. (**F**) MAPK/PI3K signaling activation in established cell lines. PC refers to positive control (0.01 mM insulin treatment). The values of densitometric measurements of western blots are added next to the bands in brackets, and denote the intensities of bands in resistant lines relative to their respective parental lines. All values were normalized to the loading control (ß-actin), and phosphorylated proteins were further normalized to total protein levels. * denotes a statistically significant difference (*p* < 0.05).

**Figure 3 biomedicines-11-01353-f003:**
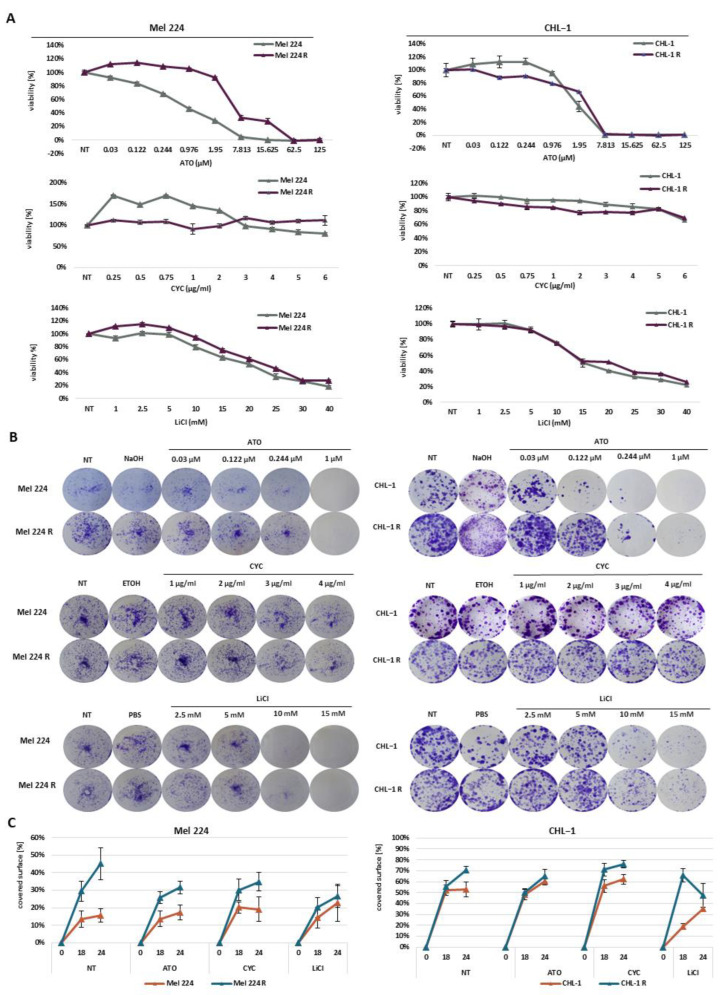
Responsiveness of established cell lines to HH-GLI inhibitors. (**A**) Cell viability examined with MTT assay after ATO, CYC, and LiCl treatment. (**B**) Colony forming capacity after ATO, CYC, and LiCl treatment. (**C**) Cell migration 18 and 24 h after ATO, CYC, and LiCl treatment. NT refers to untreated cells.

**Figure 4 biomedicines-11-01353-f004:**
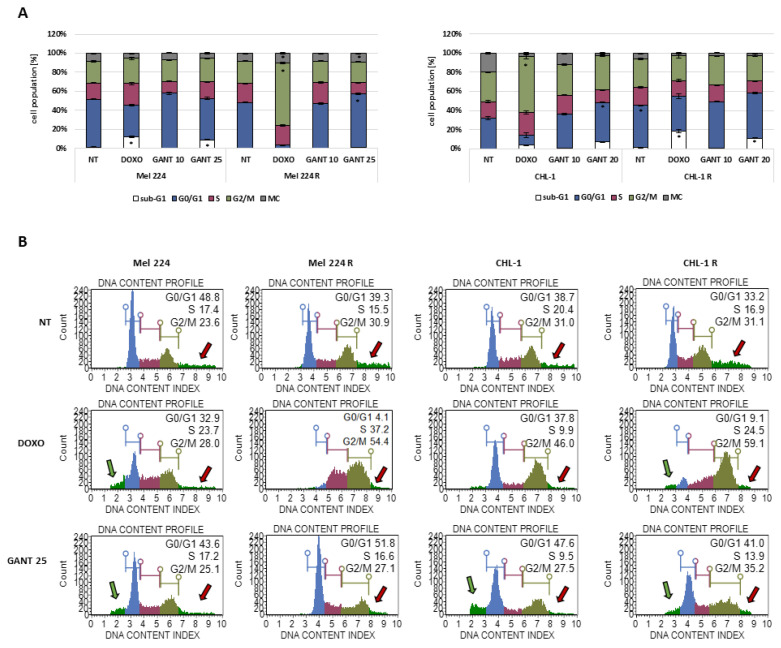
Cell cycle progression and apoptosis in established resistant cell lines. (**A**) Cell cycle analysis in untreated cells (NT) and after 100 nM DOXO, GANT-61 (10 μM and 20/25 μM) treatment. (**B**) Sub-G1 (green arrows) and multinucleated (MC) (red arrows) cell population identification after DOXO and GANT-61 treatment. * denotes a statistically significant difference (*p* < 0.05).

**Figure 5 biomedicines-11-01353-f005:**
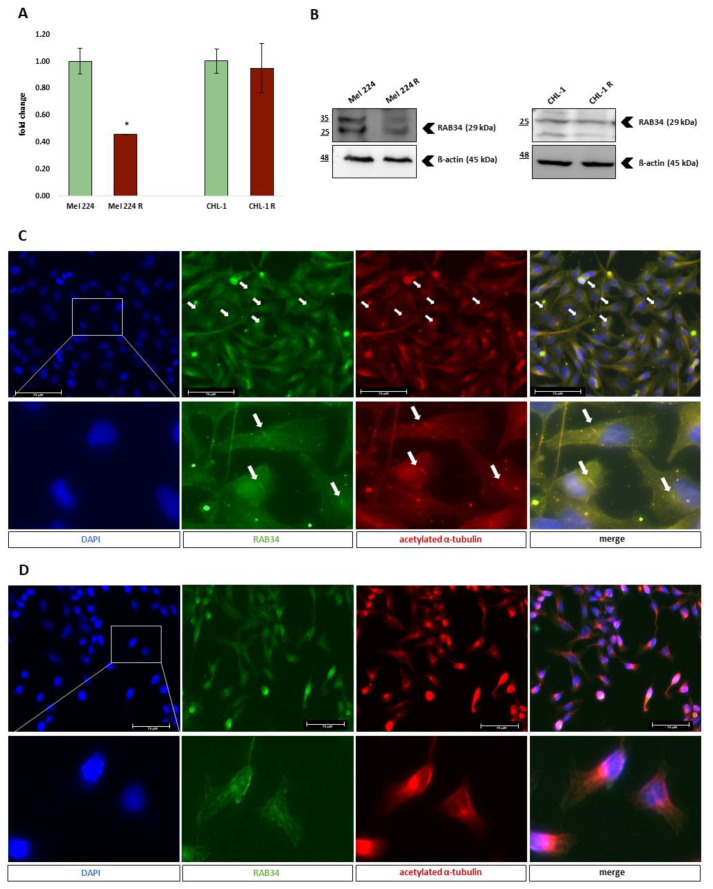
Primary cilia formation in established cell lines resistant to GANT-61. (**A**) *RAB34* gene expression evaluated by qPCR. (**B**) RAB34 protein expression evaluated by western blot. (**C**) Primary cilia (white arrows) visualization in the Mel 224 cell line. (**D**) The absence of primary cilia in the CHL-1 cell line. * denotes a statistically significant difference (*p* < 0.05).

## Data Availability

No new data were created or analyzed in this study. Data sharing is not applicable to this article.

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
