# Peer review of "Signaling Switching from Hedgehog-GLI to MAPK Signaling Potentially Serves as a Compensatory Mechanism in Melanoma Cell Lines Resistant to GANT-61"

_biomedicines, 2023, doi:10.3390/biomedicines11051353_

Round 1

Reviewer 1 Report

The reviewed article presents molecular studies of melanoma cell lines resistant to GANT-61. The Authors focused on analyzing two signaling pathways: Hedgehog-GLI and MAPK. They demonstrated, among others, that the obtained resistant cell lines were characterized by changes in cell morphology, downregulation of HH-GLI signaling components and SMO, and hyperactivation of MAPK signaling pathways. Many different laboratory techniques and analyzes were used in the study, i.e. MTT assay, Western blot, qRT-PCR, wound healing assay, colony forming assay, flow cytometry, and fluorescence microscopy. In general, the results were presented clearly and  in an aesthetic way. In my opinion, the article has great scientific potential. However, before publishing, some issues should be explained or corrected:

1. Ad. 2.2. On what basis the concentration ranges were selected?

2. Ad. 2.4. What was the concentration of phosphatase and protease inhibitors?

3. Some graphs seem to be missing statistical markings (e.g. 1B)

4. I suggest densitometric analysis for WB and presenting the results in the bar graphs.

5. Were the experiments and studies of resistant cell lines performed in the presence of GANT-61 or after a 2-week absence? I believe that the latter variant seems more reliable from the point of view of stabilized resistance and the potential direct effect of GANT-61 on cellular processes.

6. The paper lacks a paragraph summarizing the research and referring the obtained results to clinical conditions - diagnostics, prognosis, and therapy.

Author Response

1.Ad. 2.2. On what basis the concentration ranges were selected?

Answer: Thank you for your question. For MTT assay, we have tested doses that are below and over the calculated IC50 doses that were reported in literature. In one of our previous studies, we have performed MTT assay on a wide panel of melanoma cell lines and examined the effect of GANT-61, LiCl and CYC on cell viability (Kurtović et al. 2022). For other assay, such as colony forming assay, we used several concentrations below the calculated IC50 dose to “catch” a dose-dependent response, since the IC50 doses resulted in complete abrogation of colony forming ability. Cell lines used in this study showed different sensitivity to inhibitors (CHL-1 cell line was more sensitive to GANT-61 than Mel224), therefore we had to further adjust the concentration ranges. For wound healing assay, we used concentrations that were shown to be effective for one-time treatment in our previous studies.

  1. Ad. 2.4. What was the concentration of phosphatase and protease inhibitors?

Answer: Both of these products are proprietary blends of enzymes that are dissolved per manufacturer’s instructions (one tablet per 10 ml of buffer). The exact concentrations and composition of the inhibitors are not listed in the datasheets. Therefore it is impossible for us to list the exact concentrations and composition of these cocktail inhibitors in the materials and methods.

  1. Some graphs seem to be missing statistical markings (e.g. 1B)

Answer: Thank you for your comment, we added statistical markings to Figure 1B.

  1. I suggest densitometric analysis for WB and presenting the results in the bar graphs.

Answer: Thank you for the comment. The values of densitometric measurements were added to the images next to the bands, as the addition of extra graphs overcrowds the figure and makes it harder to follow.

  1. Were the experiments and studies of resistant cell lines performed in the presence of GANT-61 or after a 2-week absence? I believe that the latter variant seems more reliable from the point of view of stabilized resistance and the potential direct effect of GANT-61 on cellular processes.

Answer: Thank you for this great question.  All experiments are done on cell lines that were under constant GANT-61 treatment. Since the cell lines maintained resistance to GANT-61 after two weeks without treatment, we decided to perform all experiments on the original variant. While performing assays where cells are treated with other inhibitors, GANT-61 treatments were omitted, so we can more clearly interpret the results. Your comment is on point and we agree that it would be a great approach to perform all experiments on the second variant and to compare them with the results from the original variant which was under constant therapeutic pressure.

  1. The paper lacks a paragraph summarizing the research and referring the obtained results to clinical conditions - diagnostics, prognosis, and therapy.

Answer: Thank you for this comment. In the introduction and discussion section we mentioned the potential of targeting Hedgehog-GLI in cancer. Since GANT-61 is not suitable for clinical use, or any other selective GLI inhibitor, we could not refer our results to actual clinical conditions. Instead, we highlighted the importance and better effectiveness of targeting GLI proteins in comparison to upstream components like coreceptor SMO which often lead to therapy resistance in melanoma.

Reviewer 2 Report

In figure1, authors should add description about why both cell lines showed reduced viability at low GANT-61 concentrations.

RNAseq should be done to comprehensively see the different gene expression among Mel 224 cells, Mel 224-R cells, CHL-1 cells and CHL-1 R cells.

Authors established melanoma cell lines resistant to GANT-61, and compared some function between GANT-61 sensitive and resistant cell lines. Although some results are novel, these are overall not conclusive unfortunately. From these results, the mechanism how chemoresistance changes tumor characteristics is not known, or not suggested clearly.   In addition, different two cell lines showed different characteristics after obtaining resistance to GANT-61 in some activities, suggesting that the results are not universal and may not be reproducible.

Author Response

In figure1, authors should add description about why both cell lines showed reduced viability at low GANT-61 concentrations.

Answer: Thank you for this comment. We mentioned in the 3.1 section that reduced viability at low GANT-61 concentrations after two weeks without GANT-61 treatment could be a result of a heterogenous population in which a subset of Mel 224 R and CHL-1 R cells were not stably resistant to GANT-61 (lines 227-228).

RNAseq should be done to comprehensively see the different gene expression among Mel 224 cells, Mel 224-R cells, CHL-1 cells and CHL-1 R cells.

Answer: This is a great comment. We agree that RNA-seq analysis would be an excellent direction for further characterization of established resistant cell lines. Unfortunately, due to financial reasons, we did not plan to include RNA-seq within the framework of this study. However, in our future studies, RNA-seq analysis would be a great addition.

Authors established melanoma cell lines resistant to GANT-61, and compared some function between GANT-61 sensitive and resistant cell lines. Although some results are novel, these are overall not conclusive unfortunately. From these results, the mechanism how chemoresistance changes tumor characteristics is not known, or not suggested clearly. In addition, different two cell lines showed different characteristics after obtaining resistance to GANT-61 in some activities, suggesting that the results are not universal and may not be reproducible.

Answer: Thank you for your comment. We need to emphasize that our study brings first ever insights into GANT-61 resistant cell lines. The goal of our study was firstly to characterize established cell lines and identify molecular changes which will help us in future studies to investigate mechanisms responsible for resistance occurrence. Different characteristics of analysed cell lines were not surprising since cell lines used in our study have a different mutational background and origin. Therefore, future analysis of a broader cell line panel that covers most frequent mutations in melanoma, will give us more information about unique and specific mechanisms in this context. However, as there is no description at all of GANT-61-resistant lines in the literature, we think this work has sufficient gravity to be published even with the limited number of resistant cell lines.

Reviewer 3 Report

The authors of the manuscript „Signaling switching from Hedgehog-GLI to MAPK signaling 2 potentially serves as a compensatory mechanism in melanoma 3 cell lines resistant to GANT-61“, by Pitesa et al. etsablished melanoma cell lines with resistance against GANT-61 and characterized the cell lines. The article needs a major revision because the text is very confused and it is so hard to interpret the data.  Ultimately, the manuscript is regrettably only a description of two cell lines treated with GANT-61.

Questions/Statements:

What is the reason for not using significance stars in Figure 1?

Please adjust or correct the order of MMP9 and MMP2 for Mel224 in Figure 2C.

Figure 2 A-D is shown in a confusing way.

Here, mRNA levels of genes that are not regulated (SHH, SUFU, MMP9) could be included in a supplement. Some genes were not even discussed in the Result chapter (GSK3B).

A graphical overview of which proteins are really regulated in the same way in both cell lines is missing. A densitometry would have been helpful for this topic. Thus, it is unclear whether the experiments are really n=3 for the protein level.

In Figure 2B, GSK3beta total protein in CHL-1 is not detectable at all but the phosphorylated form should suddenly be there. Is this not unrealistic?

InFigure 2D, ERK/p-ERK should be repeated, since in MEL224 the ERK band looks so unclear.

Lane 246-248: Migration and invasion is not the same.

Lane 259: I am not sure if the phosphorylation amounts are interpreted correctly without densitometry. For both phospho ERK and phospho p38 as well as phospho JNK, the unphosphorylated form of the protein is actually the loading control in pretty much all publications of kinase data. So I think at least phospho JNK, for example, is being misinterpreted.

Figure 2F: The reason for this experiments is not explained in the result chapter and it is difficult to interprete this results.

The positive control with insulin was not explained in the result chapter. What does MAPK mean in this context. All three kinases (ERK, p38, JNK together?). What does the yelllow part „DUAL“ mean? The assay is not explained very well in the material & methods. Therefore, it is not easy to undestand all the results in Figure 2F.

Lane 276-277: U0126 treatment has not yet been mentioned in this manuscript. A misspelling of U0216 in lane 277.

It is difficult to draw a conclusion as to whether the data of Figure 2 B, D and F now fit together.  A conclusion or a summary with the essential results from 2 is also not given by the authors.

Figure 3: why did the authors use LiCl? GSK3beta was not influenced in previous experiments. The choice of inhibitors seems arbitrary. Why should ATO treatment suddenly lead to resistance (supplement).

Lane 303: What was not significantly changed in  Figure 3A and Supplementary 303 Figure S3. (The sentence is incomplete).

The different modes of action or points of intervention of the inhibitors must be listed in the Materials and Methods chapter.

Lane 312: ......higher........(higher than what?)

Figure 4 is again very confusing and a summary is missing. Lane 347: Where can the reviewer see the multinucleated cells. Are they not expected after resistance with GANT-61? Why should DOX induce resistance? Does a treatment of combination lead to apoptosis or prevention of apoptosis? A FACS analysis for apoptosis would be great.

Summary:

Unfortunately, one gets lost in the data due to the many discrepancies in the different treatments and due to the differences between protein and mRNA data and the functional experiments (e.g. cell cylce interpretation). The authors have not succeeded in creating interpretive clarity.

When you finally read the discussion, you realize that some molecules -as already mentioned by the reviewer- confuse rather than help in the results. Also in the discussion, several molecules are not taken into account. Some of these molecules can be shifted to the supplement.

The discussion is very nicely written. However, the results should be adapted or shortened more for discussion.

Author Response

What is the reason for not using significance stars in Figure 1?

Answer: Thank you for the comment, we added significance markings to Figure 1.

Please adjust or correct the order of MMP9 and MMP2 for Mel224 in Figure 2C.

Answer: Thank you for noticing this. We corrected the order of MMP9 and MMP2 in Figure 2C.

Figure 2 A-D is shown in a confusing way.

Here, mRNA levels of genes that are not regulated (SHH, SUFU, MMP9) could be included in a supplement. Some genes were not even discussed in the Result chapter (GSK3B).

Answer: Thank you for the comment. This figure was an attempt to measure the detected changes in the gene/protein expression profiles as comprehensively as possible. Therefore, it is not surprising that some of the genes/proteins showed no changes in their expression. Still, we would like to keep the complete panel in the figure as it demonstrates that not all signaling components change in the resistant lines, and this is also relevant information for future research in this area. Hiding the non-responsive genes/proteins in the supplement would result in an incomplete image.

A graphical overview of which proteins are really regulated in the same way in both cell lines is missing. A densitometry would have been helpful for this topic. Thus, it is unclear whether the experiments are really n=3 for the protein level.

Answer: Thank you for the comment. The values of densitometric measurements were added to the images next to the bands, but addition of extra graphs overcrowds the figure and makes it harder to follow.

In Figure 2B, GSK3beta total protein in CHL-1 is not detectable at all but the phosphorylated form should suddenly be there. Is this not unrealistic?

Answer: Thank you for pointing this out. We agree with your comment and were surprised with this observation. We believe that variable antibody quality probably led to this mixed result. However, we decided to present the data as is rather than excluding GSK3B findings, even though it would make a “prettier” image.

In Figure 2D, ERK/p-ERK should be repeated, since in MEL224 the ERK band looks so unclear.

Answer: Thank you for the comment. We replaced the ERK 1/2 blot with a new one which is clearer. We believe that the pERK 1/2 band looks clear enough so the readers can see that the Mel 224 cell lines has lower levels of phosphorylated ERK than Mel 224 R.

Lane 246-248: Migration and invasion is not the same.

Answer: Unfortunately, we could not identify the part you are referring to, since in our manuscript version lanes 246-248 are empty.

Lane 259: I am not sure if the phosphorylation amounts are interpreted correctly without densitometry. For both phospho ERK and phospho p38 as well as phospho JNK, the unphosphorylated form of the protein is actually the loading control in pretty much all publications of kinase data. So I think at least phospho JNK, for example, is being misinterpreted.

Answer: We have re-evaluated the phospho-ERK, -JNK and p38 values by two-step densitometry. First, we normalized all the intensities to the loading control, and then calculated the phospho-to-total protein ratio for each of the three proteins in question. We are very grateful for your assistance as it has slightly altered the results regarding phospho-JNK, as you yourself noticed. We added the densitometry values to Figure 2 and appropriate descriptions to the results and discussion sections.

Figure 2F: The reason for this experiments is not explained in the result chapter and it is difficult to interprete this results.

Answer: The reason why we performed flow cytometry was to quantify the trend that was previously detected with western blot analysis in the resistant cell lines regarding MAPK signaling hyperactivation. In case of Mel 224 cell line, we showed that acute GANT-61 treatment induces higher ERK activity in the Mel 224 R compared to Mel 224, while in CHL-1 cell line, we did not observe any changes. The flow cytometry kit that we used measures p-ERK 1/2 and p-AKT, so it could not detect p-p38 that was dramatically changed in CHL-1 R cell lines.

The positive control with insulin was not explained in the result chapter. What does MAPK mean in this context. All three kinases (ERK, p38, JNK together?). What does the yelllow part „DUAL“mean? The assay is not explained very well in the material & methods. Therefore, it is not easy to undestand all the results in Figure 2F.

Answer: Thank you for pointing this out. We have added additional details in the Methods section which will help readers to better understand the setup of this experiment. The positive control, the insulin treatment ensures a basal detectable level of MAPK activity, since it is known that insulin serves as a ligand for MAPK and PI3K signaling pathways. As we mentioned in the previous question, the kit that we used for flow cytometry measures the dual activity of MAPK and PI3K signaling pathways, namely p-ERK1/2 and p-AKT. By using this kit, we can detect the cell populations with activated MAPK signaling pathway (red segments in Fig.2F), activated PI3K (teal segments in Fig.2F) or both active signaling pathways (marked as DUAL, yellow segments in Fig.2F) This kit cannot measure the activity of other signaling components of MAPK, such as p-JNK and p-p38.

Lane 276-277: U0126 treatment has not yet been mentioned in this manuscript. A misspelling of U0216 in lane 277.

Answer: Thank you for noticing, we corrected the misspelling. We did mention in the 3.2 section that U0126 was used as a MAPK inhibitor and that U0126 treatment increased the MAPK/PI3K negative population in Mel 224 R cells (that showed higher MAPK activity after GANT-61 treatment) much more than in Mel 224 cells. In the case of CHL-1 cell line, we did not detect any changes after U0126 treatment.

It is difficult to draw a conclusion as to whether the data of Figure 2 B, D and F now fit together.  A conclusion or a summary with the essential results from 2 is also not given by the authors.

Answer: Thank you for this comment. We believe that the obtained results from Figures 2 B, D and F draw to a specific conclusion, and we did summarize these results in the discussion section. In Figure 2B we demonstrated that both resistant cell lines exhibited downregulation of specific HH-GLI components (GLI1, GLI2, GLI3, PTCH1 and SMO). Downregulation of HH-GLI can be an adaptive mechanism to avoid sensitivity to GANT-61 treatments. In Figure 2D we reported upregulated MAPK signaling in the resistant cell lines, in the case of Mel 224 R cells, upregulated ERK and JNK, and in case of CHL-1 R cells, p38 pathway. MAPK hyperactivation can potentially serve as compensatory mechanism for the maintenance of tumorigenic cell properties in conditions of HH-GLI downregulation. In Figure 2F which includes flow cytometry results, we confirmed and quantified western blot results for the Mel 224 cells. We demonstrated that Mel 224 R cells upregulate more pERK levels after GANT-61 treatment than the parental cell line and show higher sensitivity to MEK inhibitor, U0126 in terms of MAPK activity. Due to the specificity of antibodies used in this kit, we could not analyse p38 activity in CHL-1 R cells which have shown upregulation of p38.

Figure 3: why did the authors use LiCl? GSK3beta was not influenced in previous experiments. The choice of inhibitors seems arbitrary. Why should ATO treatment suddenly lead to resistance (supplement).

Answer: We wanted to include several known HH-GLI inhibitors as possible in this study, so we can characterize established cell lines in more detail regarding potential cross-resistance. In our study, we included inhibitors that that are regularly used in other studies and cover major HH-GLI components, upstream ones like SMO and GSK3ß and downstream like GLI proteins. Since GSK3ß is important for GLI protein processing, we wanted to examine the effect of lithium chloride on different levels. Unfortunately, GSK3ß influences many signaling pathways besides HH-GLI, so we could not draw a clear conclusion on how lithium chloride mechanistically affects our cells.

The established Mel 224 R cell line showed transient resistance to ATO which acts as an unspecific GLI protein antagonist. We believe that these cells exhibited drug tolerance (as a result of these unspecific characteristics) since the effect was reversed after two weeks without GANT-61 treatment. Continuous GANT-61 treatments caused and probably enhanced this effect, but due to other targets that ATO affects, the effect was lost once GANT-61 treatments were removed. In this case we concluded that continuous GANT-61 treatment leads to ATO tolerance, but once GANT-61 is removed, the cells become sensitive again due to the unspecific traits of ATO.

Lane 303: What was not significantly changed in Figure 3A and Supplementary 303 Figure S3. (The sentence is incomplete).

Answer: We could not identify the missing part in this sentence. It is clearly written that the effect of tested inhibitors: SMO (CYC, SDG and VDG) and RAS (SAL) on cell viability was not statistically changed (Figure 3A and Suppl. Figure 3).

The different modes of action or points of intervention of the inhibitors must be listed in the Materials and Methods chapter.

Answer: Thank you for this comment. We did mention how each inhibitor acts in the Discussion part, but now we have also added these details in the Methods section so the readers can more clearly interpret the results.

Lane 312: ......higher........(higher than what?)

Answer: Thank you for pointing this detail. We added in this sentence that the colony formation capacity in resistant cell lines was higher compared to parental cell lines, so it is clearer.

Figure 4 is again very confusing and a summary is missing. Lane 347: Where can the reviewer see the multinucleated cells. Are they not expected after resistance with GANT-61? Why should DOX induce resistance? Does a treatment of combination lead to apoptosis or prevention of apoptosis? A FACS analysis for apoptosis would be great.

Answer: Thank you for your comment. We believe that Figure 4. shows cell cycle analysis results in a clear manner (summarized in a stacked chart rather than many individual charts). Firstly, in Figure 4A, cell cycle analysis is showed graphically where you can see identified cell populations: sub-g1, G0/G1, G2/M and multinucleated cells (MC), and in Figure 4B the same results are presented color-matched with histograms, so the readers can see peaks for each cell population. We did summarize these results in the discussion section.

Multinucleated cells (MC) can be identified as a peak after G2/M population. We highlighted MC peaks in Figure 4B with red arrows. Besides that, we identified multinucleated cells using immunofluorescence, but we did not include these images in the manuscript since we already quantified MC population with flow cytometry. We have now added to the supplementary materials (Figure 4S) an example of identified multinucleated cells, so the readers can have a clearer picture of the cell cycle experiment. In the 3.4 result section and discussion part we mentioned the importance of MC in cancer through genomic instability and resistant cell traits. In our case, MC were present in all samples, regardless of GANT-61 resistance, so we did not connect MC occurrence with therapy resistance.

DOXO treatment was used only as a positive control for G2/M cell cycle arrest. DOXO was not connected in any way to resistance induction in our study. We have shown that in comparison to Mel 224 R, DOXO induces sub-G1 population in Mel 224 cell line which can indicate apoptosis. On contrary, CHL-1 R cell line showed the same effect after DOXO treatment in comparison to CHL-1. We believe that this results can indicate apoptosis or slow-cycling cell state. For that reason, we agree, apoptosis analysis would be a great addition to clarify these observations. It would be possible to provide this, but it would require at least 2-3-month extension, and not the 10 days we were given.

For cell cycle analysis, we used single DOXO and GANT-61 treatments. Combination treatments were not performed, so it is hard to speculate how combined treatment would influence apoptosis (occurrence of sub-G1 population).

Summary:

Unfortunately, one gets lost in the data due to the many discrepancies in the different treatments and due to the differences between protein and mRNA data and the functional experiments (e.g. cell cylce interpretation). The authors have not succeeded in creating interpretive clarity.

When you finally read the discussion, you realize that some molecules -as already mentioned by the reviewer- confuse rather than help in the results. Also in the discussion, several molecules are not taken into account. Some of these molecules can be shifted to the supplement.

The discussion is very nicely written. However, the results should be adapted or shortened more for discussion.

Answer: Thank you for the summary. It would actually benefit the manuscript to present results and discussion in a combined way. It would make it easier to follow investigative reasoning and to discuss each point as it comes. It is hard to find the balance between too descriptive and not descriptive enough, and that is why we focused on the discussion to put everything into context.

Round 2

Reviewer 1 Report

The manuscript has been improved. The Authors have answered the questions and considered the remarks. Thus, I recommend publishing the article. 

Reviewer 2 Report

Authors modified based on the comments from reviewers, and I have no more comments.

Reviewer 3 Report

Dear Reviewer,

thank you for correction.

It‘ s acceptable, now.